# Qualitative study to identify ethnicity-specific perceptions of and barriers to asthma management in South Asian and White British children with asthma

Monica Lakhanpaul,[1] Lorraine Culley,[2] Tausif Huq,[3] Deborah Bird,[4] Nicky Hudson,[2] Noelle Robertson,[5] Melanie McFeeters,[6] Logan Manikam,[7] Narynder Johal,[8] Charlotte Hamlyn-Williams,[1] Mark R D Johnson[9]

For numbered affiliations see end of article.

**Correspondence to**
Professor Monica Lakhanpaul;
m.lakhanpaul@ucl.ac.uk

## ABSTRACT

**Objective** This paper draws on the data from the Management and Interventions for Asthma (MIA) study to explore the perceptions and experiences of asthma in British South Asian children using semi-structured interviews. A comparable cohort of White British children was recruited to identify whether any emerging themes were subject to variation between the two groups so that generic and ethnicity-specific themes could be identified for future tailored intervention programmes for South Asian children with asthma.

**Setting** South Asian and White British children with asthma took part in semi-structured interviews in Leicester, UK.

**Participants** Thirty three South Asian and 14 White British children with asthma and aged 5–12 years were interviewed.

**Results** Both similar and contrasting themes emerged from the semi-structured interviews. Interviews revealed considerable similarities in the experience of asthma between the South Asian and White British children, including the lack of understanding of asthma (often confusing trigger with cause), lack of holistic discussions with healthcare professionals (HCPs), an overall neutral or positive experience of interactions with HCPs, the role of the family in children's self-management and the positive role of school and friends. Issues pertinent to South Asian children related to a higher likelihood of feeling embarrassed and attributing physical activity to being a trigger for asthma symptoms.

**Conclusions** The two ethnicity-specific factors revealed by the interviews are significant in children's self-management of asthma and therefore, indicate the need for a tailored intervention in South Asian children.

## INTRODUCTION

Asthma is one of the most prevalent chronic conditions in childhood. Estimates in the UK suggest 1.1 million, or one in ten children, will experience asthma at some point in their childhood.[1] Evidence show that children from South Asian communities experience under-recognition of asthma symptoms, worse

## Strengths and limitations of this study

► An unequal power relationship between adult researchers and child participants has been mitigated through the use of 'child-friendly' data collection methods.
► The study was not resourced to analyse children's drawings, although if done systematically, this may provide additional useful insights into children's understanding and experience of asthma.
► Some children declined to participate in the activities provided, such as drawing, and the activities themselves increased the length of interview time.

disease severity and increased attendance at the emergency department, exceeding the rates of their White British counterparts.[2–4] However, there is no evidence to indicate South Asian children experience more severe asthma or have a genetic predisposition to severe asthma attacks.[2–4]

Barriers to effective asthma management in South Asian children are well-recognised in literature.[2–4] However, many health interventions aimed at mitigating these barriers have failed to improve asthma outcomes, often ascribed to a lack of cultural sensitivity and a failure to recognise the ethno-religious heterogeneity within the South Asian community.[5] A systematic review by Lakhanpaul et al identified several cultural and structural barriers to optimal asthma management in South Asian children, including diverse knowledge of asthma, non-acceptance and parental denial of asthma.[6] It highlighted the importance of identifying the nature and mechanism of these barriers, so that ethnically-tailored interventions could be developed. 'Tailored' interventions consider individual cultural influences and are more effective

than 'modified' interventions, which modify an intervention designed for a majority population.[6 7]

There is also a remarkable under-representation of UK minority-ethnic children in all aspects of asthma research; thus, specific ethnicity-specific and religion-specific factors are rarely integrated into health services.[8] This is despite international (UN Convention on the Rights of the Child[9]) and national policy mandates (including the Children Act 1989 and 2004 affecting England and Wales) to increase precedence of the views and opinions of children and young people (CYP). As CYP are not a homogenous group and have varying health needs depending on their backgrounds, capabilities and interests, it is important that their views are heard, especially in childhood asthma.

## Context of the current study: the MIA project

This paper is drawn from the Management and Intervention for Asthma (MIA) study carried out in an urban area in Leicester, UK. It had an overall aim of developing an intervention-planning framework for British South Asian children with asthma, guided by a participatory collaborative method which involved individuals and communities defining problems and developing solutions.[10]

This paper reports on one of the major phases of the MIA study: exploring the perceptions and experiences of asthma in British South Asian children by semi-structured interviews. A comparable cohort of White British children was recruited to identify whether any emerging themes were subject to variation between the two groups, so that generic and ethnicity-specific themes could be identified for future tailored intervention programmes in British South Asian children with asthma. Children's interviews were complemented by interviews with parents/carers, community members and healthcare professionals (HCPs), the results of which are discussed elsewhere.[10]

## METHODS

This study adopted a qualitative methodology using semi-structured interviews.

## Participant sampling

Families with children aged 5–12 years with a diagnosis of asthma were included (table 1). Participants were purposively sampled via two routes: via healthcare (eg, general practice) and the community, facilitated by community facilitators , who were bilingual members of the South Asian community trained as lay researchers to assist with the study.

Recruitment ensured a proportional representation of six major South Asian ethno-religious groups (table 1; these terms describe ethno-religious identity and not participants' nationality), with parents asked to assign children's demographic information. From the recruited families, purposive sampling was used to recruit children (matching age, sex and asthma severity according to British Thoracic Society guidelines (BTS)[10]). In some

| Table 1 Demographic information of semi-structured interview participants | South Asian children (%) | White British children (%) |
|---|---|---|
| Total number | 33 | 14 |
| Male | 20 (61) | 8 (57) |
| Female | 13 (39) | 6 (43) |
| BTS grading | | |
| Level 1 | 7 (21) | 3 (21) |
| Level 2 | 17 (52) | 8 (57) |
| Level 3 | 6 (18) | 3 (21) |
| Level 4 | 3 (9) | 0 |
| Ethno-religious/ethnic group | | |
| Indian Gujarati Hindu | 6 (18) | – |
| Indian Gujarati Muslim | 4 (12) | – |
| Indian Gujarati Jain | 1 (3) | – |
| Indian Punjabi Sikh | 6 (18) | – |
| Indian Punjabi Hindu | 1 (3) | – |
| Bangladeshi Muslim | 9 (27) | – |
| Pakistani Muslim | 6 (18) | – |
| White British | – | 13 (93) |
| White British (English) | | 1 (7) |
| White British (Wales) | – | 0 |
| White British (Scottish) | – | 0 |
| White British (Northern Irish) | – | 0 |
| Religion | | |
| Hindu | 7 (21) | 0 |
| Muslim | 19 (58) | 0 |
| Sikh | 6 (18) | 0 |
| Jain | 1 (3) | 0 |
| Christian | 0 | 6 (43) |
| Atheist | 0 | 1 (7) |
| No religious belief | 0 | 4 (29) |
| Not specified | 0 | 3 (21) |
| Age | | |
| 5 years old | 2 (6) | 2 (14) |
| 6 years old | 5 (15) | 2 (14) |
| 7 years old | 5 (15) | 0 |
| 8 years old | 4 (12) | 4 (29) |
| 9 years old | 3 (9) | 1 (7) |
| 10 years old | 5 (15) | 4 (29) |
| 11 years old | 5 (15) | 1 (7) |
| 12 years old | 3 (9) | 0 |
| Not specified | 1 (3) | 0 |
| Parent/carer highest level of education achieved * | | |
| Primary school | 4 (8) | 0 |
| Secondary school | 21 (43) | 11 (37) |

Continued

**Table 1** Continued

|  | South Asian children (%) | White British children (%) |
| --- | --- | --- |
| College education | 3 (6) | 3 (10) |
| University | 20 (41) | 14 (47) |
| Not specified | 1 (2) | 2 (7) |

*Numbers may not be equal to the total number of children interviewed as either one or both parents responded to the questionnaire.
BTS, British Thoracic Society.

instances, more than one child was recruited from a family.

Thirty three South Asian children and 14 White children were interviewed. BTS grading allowed a wide range of asthma experiences to be captured (table 1).

### Consent and assent
Assessment of 'Gillick competency' was used to obtain consent from children aged 10–12 years.[11] Younger children were judged by the research team on their ability to provide assent orally where appropriate.

### Data collection
A question schedule was developed from the findings of an initial scoping review by external advisors (online supplementary file 1).[10] Children also produced drawings and text during the interviews which guided interview questions. Interviews were carried out by two female researchers (a paediatrician and a social scientist, both trained in qualitative methods and not involved in their care). Interviews took place in the children's family homes with some choosing to give interviews with their parents. Interviews were digitally recorded, transcribed and field notes were taken. No repeat interviews were conducted.

### Data analysis
Interview transcripts were analysed according to the principles of interpretive thematic analysis using NVivo (QSR International, Warrington, UK).[10] Transcripts were not returned to children for comments/corrections.

A detailed explanation of the thematic analysis can be found in previous publications.[10 12] Briefly, two analysts independently analysed the transcripts to create a framework of thematic categories from both interviews. The thematic frameworks of South Asian and White British children's data were then closely compared and amended to ensure consistency of meaning across both datasets to create a comprehensive coding framework. All transcripts were then systematically coded using the resultant framework.

### Patient and public involvement
Children were not directly involved in the design, recruitment or conduct of the study but participated in the development of an intervention framework following the interviews in a series of community-based workshops.[10] Parents were included as members of the research team (NJ, as co-applicant) and as members of the advisory group. Finally, a number of dissemination events were held in the community involving South Asian and White British children.

## RESULTS
The key themes arising from the exploration of the interviews were: understanding of asthma; children's self-management of asthma; perception of asthma; management in school; the role of friends; and interactions with healthcare providers.

### Children's understanding of asthma
Fourteen South Asian children (42%) and six White British children (43%) explained what they thought to be the cause for asthma, although most were uncertain. Some children attributed causes of asthma to triggers such as the weather, the environment, or viruses.

> Well I think it's where if you are near something that triggers it off your lungs might tighten together a little bit and stop you breathing and lock up your air pipe a little bit so you will get quite wheezy (White British boy)

> As mum keeps saying over and over again, the damp weather, […] I may have caught asthma from the damp weather… (Indian girl)

Two South Asian and two White British children thought that asthma had a hereditary component.

> I know it runs in families and stuff because me and my mum have got it. (White British girl)

> Probably because like someone in their family before like your grandma or grandfather had it […] (Punjabi boy)

### Children's interactions with the healthcare system
Seven South Asian (21%) and seven White British children (50%) could not recall the time they were diagnosed with asthma. Furthermore, children were more likely to remember their recent interactions with HCPs rather than those in the past.

Overall, the children's interactions with HCPs were described broadly as positive or neutral. Children were more likely to recall their visits to their general practitioner (GP) since most had visited them recently. Positive comments focused on their GP's friendly approach and communication, including explaining asthma and medications. Three negative comments related to 'horrible tasting tablets', misdiagnosis of hay fever and being 'ignored' after initiating nebuliser.

> He [doctor], he'll tell you what's happening, what's wrong, and if you have asthma or if you don't have

asthma, and he'll tell you if you need a, the stuff for the inhaler you when you put it in. (Pakistani boy)

In both groups, experiences and perceptions of secondary care were generally less positive, often involving negative emotional descriptions and related to being treated for acute asthma attacks. Nevertheless, the quality of interactions with individual HCPs remained positive.

Just a little bit scary. […] Because sometimes you have to have injections. [The nurses are] so kind. […] So what [the doctors] do injections they, when they've done it, [the nurses] cheer me up when I cry. (Indian girl)

South Asian and White British children did not recall having a holistic discussion with the HCPs such as discussions about school, physical activity and quality of life. Some also recalled GPs not involving children during consultations.

[Doctors should talk to] both [me and my parents] […] Like, to explain […] what is it happening, […] after explaining to the parents[…] (Bangladeshi girl)

### Children's self-management of asthma

Twenty nine South Asian (89%) and four White British children (29%) spoke about self-assessing symptoms and self-management, especially as they became older. In both groups, parents were described as being responsible for reminding and instructing children to take inhalers and facilitating the process of self-management.

The families' response to an asthma attack appeared to mediate a child's symptom perception. Both groups of children reported that parents, especially their mothers, played a vital role in managing acute attacks by spotting symptoms and encouraging inhaler use.

My mum tells me […] how many puffs […] when she wants to do something else, […]she.she.she says how many times to do it. And then I start doing it. […]Sometimes my dad helps me when I'm like stuck. (Indian girl)

It was evident that the older children devised their own system of self-management, with the help of their parents.

…so always have two inhalers of the same colour, like… if this one ran out then I'd tell my Mum or Dad or sister… (Indian boy)

Overall, parents (especially mothers) in both groups of children were felt to play an important role in mediating children's symptom perception and self-management of routine and acute attacks, with no specific ethnic differences.

### Children's perception of the impact of asthma

One of the biggest impacts of asthma on children as reported was how it affected their ability to participate in sporting activities and physical education classes at school. South Asian children more frequently attributed physical activity to be a trigger for an asthma attack (n=29, 89%) than White British children (n=7, 50%). Common symptoms included breathlessness and coughing, resulting in children either sitting out or avoiding activities altogether.

Because you know when I run the wheeziness starts and make me cough […] So that's why I take the inhaler sometimes when we come back from sports or anything. (Indian boy)

*I have had to do it once (take inhaler) before but it was sports day […]. (White British girl)*

The most commonly reported emotions were embarrassment (n=11 South Asian; n=3 White British), fear (n=3 South Asian; n=2 White British), upset (n=7 South Asian; n=1 White British) and unfairness (n=2 South Asian; n=2 White British). South Asian children were more likely to report embarrassment in relation to sports, school and taking medication than White British children. It was described, however, with some emotional distance, as an embarrassing condition rather than children being embarrassed (See *School and friends*).

[…] she's grown up, she says, It's embarrassing to carry it [inhaler], mum. (Mother of a Bangladeshi girl)

That I have to cough when my sister doesn't cough, or I have to have more medication than she has to have. It's a bit annoying sometimes but you get used to it. (White British girl)

Both groups of children expressed few positive emotions and when asked if they had one wish, many reported that they wanted asthma to 'go away'. A few children did find some positives in their experience of asthma management.

The good thing is erm […] I think I've improved in my confidence and my determination […]. (Bangladeshi boy)

### School and friends

Peers and friends were described as having an important role in children's experience and management of asthma. Having friends who understood their asthma, and in some cases, could help if needed, was deemed important for both South Asian and White British children.

At school it doesn't affect me much, people don't tease me or anything…, their friends are aware of my asthma, some of them have asthma themselves and they help me. (Pakistani boy)

… they [friends] mostly help me, like they usually go get it [inhaler] for me. (White British girl)

Asthma was considered a reason to feel different or 'not normal', thus a small number of children described strategies to manage the amount of information they disclosed to their peers.

I have to explain it a lot. . . they understand I have got it and I think they are alright because they know it's not catching. (White British girl)

One White British girl felt that using her inhaler at school attracted unwanted attention and one South Asian child reported being teased about his asthma.

## Suggestions for service improvements

Some children discussed ways in which knowledge of asthma could be improved and offered ideas for future interventions. Children reported wanting to speak to someone who had experienced asthma, either living with asthma or a HCP. Children also described ways in which information can be delivered.

Pictures maybe. (Indian boy)

A presentation[. . .]. They tell you things that I did not know like in more detail.[. . .] The chemist and the GP were ok but the school one explained it best and easy to understand. (Indian boy)

They [school] could do a little show about asthma or something like that […] (White British girl)

## DISCUSSION

Semi-structured interviews with children in this study provided an opportunity for those affected by asthma to offer their experiences and perceptions of asthma management. The interviews revealed that some perceived challenges facing children with asthma may be common regardless of ethnicity. However, certain experiences appeared to be more relevant to South Asian children, in particular, feeling of embarrassment and attributing physical activity to being a trigger for asthma symptoms.

Children as young as 6 years old are able to self-manage asthma with parents having a positive intermediary effect.[13] Therefore, knowledge of asthma will support and nurture the practice of self-management. However, both groups of children expressed confusion and misconceptions about the causes, triggers, and nature of asthma.[14 15] These findings indicate that the HCPs may not be adequately delivering health information and/or checking the understanding of those receiving the information. Consequently, the need for an improved information provision is echoed by the children in the suggestions they offered for improved asthma support.

Interactions with HCPs were mostly reported as positive or neutral by the South Asian and White British children and did not reveal any ethnicity-specific barriers to communication. Some children described being 'left out' during doctors' consultations which were often directed at their parents. Children often recalled providing a clinical history and being examined but did not recollect having any holistic discussions. This is consistent with the data collected from an earlier phase of the MIA study[10] which found that HCPs tended to focus on addressing medications with limited discussions about holistic management. However, the lack of recall of interactions with HCPs may also relate to children's age and the time of their recent interaction.

Physical activity is recommended within asthma management guidelines.[16] Our results show that South Asian children are more likely to attribute physical activity to being a trigger for asthma symptoms than White British children. Indeed, it has been reported that South Asian children are 3.6 times more likely to suffer from exercise-induced bronchoconstriction than White British children.[17] Regardless of the underlying mechanism, our results may indicate the need for a tailored approach to change the perceptions of physical activity in South Asian children.

A desire to be normal was commonly expressed by the children, echoing previous studies examining the psychosocial aspects of asthma.[18 19] The children struggled to talk positively about asthma, with negative emotions equally apparent in both groups. However, the self-conscious emotion of embarrassment was more likely to be experienced by South Asian children, and often related to day-to-day activities such as sports or taking medication when the children were seen as visibly different, rather than having asthma per se. However, this must be interpreted cautiously as there were more 'older' children (11–12 year old) in the South Asian cohort compared with White British children, and this perception may be a reflection of the age difference. Nevertheless, the sense of embarrassment may be compounded by an indirect experience of asthma-related stigmatisation ('courtesy stigma'[20] from the wider South Asian community.[10 12] Children who feel embarrassed are less likely to carry or use inhalers with an overall negative impact on self-management.[18] Therefore, a tailored intervention to tackle emotional effects in South Asian children should be considered.

## Challenges and limitations

Semi-structured interviews with children can involve unequal power dynamics between an adult researcher and the child participant, with differing power dynamics at play when interviewed in or without the presence of parents.[21 22] Efforts were made to minimise this impact by using child-friendly strategies.[10] The children's age range (5–12 years) is also a limiting factor (table 1), potentially influencing asthma experience and recall. Furthermore, interviewing young children also increases costs, both in terms of resources such as child-friendly interview/involvement resources, and due to increased time needed to establish trust and maintain interest.

## POLICY PRACTICE AND CONCLUSION

Interviews with children from South Asian and White British communities revealed considerable similarities in their experiences and perceptions of asthma and its management. It is therefore important to work with both South Asian and White British children to design interventions which take into consideration the important common issues raised in this qualitative study. However, two ethnicity-specific themes emerged from the interviews, which suggest that there may also be a need for an intervention

to include a more tailored component for South Asian children, particularly around physical activity and social stigma. The MIA study also included interviews and focus groups with parents/carers, HCPs and wider community members, revealing further ethnicity-specific barriers and facilitators to asthma management in South Asian children which can indirectly affect children's experience.[10] Overall, these data can contribute to the development of a multifaceted tailored intervention to improve the management of asthma in South Asian children.

**Author affiliations**
[1]Population, Policy and Practice, University College London Institute of Child Health, London, UK
[2]School of Applied Social Sciences, De Montfort University, Leicester, UK
[3]School of Medical Education, King's College London, Guy's King's and St Thomas' Hospital, London, UK
[4]Ealing Community Paediatric Service, Ealing Child Development Team, London, UK
[5]Department of Clinical Psychology, Centre for Medical Humanities, University of Leicester, Leicester, UK
[6]NHS England, Specialised Commissioning East Midlands, Leicestershire, UK
[7]Department of Epidemiology and Public Health, University College London Institute of Epidemiology and Health Care, London, UK
[8]Parent representative, Leicester, UK
[9]Mary Seacole Research Centre, DeMonfort University, Leicester, UK

**Acknowledgements** The authors would like to express their thanks to the following: All families, children, community members and healthcare professionals who participated. The community facilitators and advisory committees.* Professor Jonathan Grigg who contributed to the conception and design of the study and gave advice throughout the project. External advisor, Professor Mike Thomas. Maya Lakhanpaul, Chelsea Huddlestone and Aadil Ali who were our youth facilitators. Ms Gill Perkins and Jo Wilson who assisted with the reviewing of the literature for the systematic evidence synthesis in phase 1. Mrs Stephanie Langan, Mrs Elaine Huddlestone and Mrs Liz Knight. Mrs Emma L. Angell for coding and summarising interview data. Ms Emma Alexander for her thoughtful comments on the draft manuscript. University of Leicester, Leicester Royal Infirmary and the Children's Assessment Unit, Leicester Community Child Health Services, all participating community centres, Mosques and Gurdwaras, organisations, creches*, LCEHR. Asthma UK, with particular thanks to Brigid Hall and Leanne Metcalf. The PCRN and CLRN and the Leicester Community Children's Partnership. Dr Jo Forster. Dr Neena Lakhani. Dr Marian Carey. De Montfort University. Staff at our recruitment sites.* This research was supported by the National Institute for Health Research (NIHR) Collaboration for Leadership in Applied Health Research and Care North Thames at Bart's Health NHS Trust (NIHR CLAHRC North Thames). The views expressed in this article are those of the author(s) and not necessarily those of the NHS, the NIHR, or the Department of Health and Social Care.

**Contributors** ML, DB, LC, NR, NJ, MMc, MRDJ made substantial contributions to the conception and design of the study. ML was responsible for the overall direction of the project. ML, NR, DB, LC, NH, NJ, MM, CH-W, MRDJ contributed to the analysis and interpretation of data. ML, DB, LC, NR, NJ, MMc, MRDJ contributed to the systematic evidence synthesis. ML, DB, LC, NR, NJ, MMc designed and delivered the participatory workshops. ML, DB, LC, NR, NH, NJ, MMc, CH-W, LM, TH, MRDJ contributed to the drafting of the report and have given final approval of this report.

**Funding** This project was funded by the National Institute for Health Research. Health Services and Delivery Research (HS&DR) programme (project number 09/2001/19). NIHR had no role in the design, analysis or writing of this article. Logan Manikam is funded by a National Institute for Health Research (NIHR) Doctoral Research Fellowship (DRF-2014-07-005). Monica Lakhanpaul is funded by the NIHR Collaboration for Leadership in Applied Health Research and Care (CLAHRC) North Thames at Bart's Health NHS Trust.

**Competing interests** During the MIA project, Monica Lakhanpaul was appointed as a National Institute for Health and Care Excellence (NICE) Fellow, member of the NHS Evidence advisory board, the Health Technology Assessment advisory panel and the Drugs and Therapeutics Bulletin editorial panel.

**Patient consent for publication** Obtained.

**Provenance and peer review** Not commissioned; externally peer reviewed.

**Data sharing statement** No additional data are available.

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
