## [Reviewer comments · BMJ Open]

ARTICLE DETAILS

TITLE (PROVISIONAL)	A qualitative study to identify ethnic-specific perceptions of and barriers to asthma management in South Asian and White British children with asthma
AUTHORS	Lakhanpaul, Monica; Culley, Lorraine; Huq, Tausif; Bird, Deborah; Hudson, Nicky; Robertson, Noelle; McFeeters, Melaine; Manikam, Logan; Johal, Narynder; Hamlyn-Williams, Charlotte; Johnson, Mark

VERSION 1 – REVIEW

REVIEWER	Dr Darren Sharpe University of East London Colleagues with the lead author via the Children and Young Peoples theme, which forms part of the NIHR North Thames CLAHRC.
REVIEW RETURNED	23-Jun-2018

GENERAL COMMENTS	The redraft is much improved and its contribution to understanding the issues for health service design for children and young people from South Asian backgrounds is very and insights can be transferred to other service areas treating chronic conditions in childhood.
---

REVIEWER	Dhenuka Radhakrishnan Children's Hospital of Eastern Ontario Ottawa, Canada
REVIEW RETURNED	29-Jun-2018

GENERAL COMMENTS	This manuscript is a significant improvement over the previous. All prior reviewer concerns have been well addressed. The inclusion of many examples of specific responses from the study participants enhances the interest and usefulness of this paper both for practicing clinicians and researchers. I have only a few minor comments: 1. In the results section titled "Children's interactions with healthcare system" - line 239 states that the lack of recall of children regarding the time they were diagnosed with asthma "may reflect the quality of the interaction they had had with the HCP's." Is there any evidence to support this interpretation? I would suggest removing this statement unless there is further evidence within the study or published literature to support it. Furthermore, this statement, as well as the statement in line 241, "suggesting a lack of recall may be age-and time-dependent" are interpretations and should be included in the discussion, rather than results section of the manuscript.
---

	2. I note that there are significantly fewer 11 and 12 year olds (i.e. older children) among the White British children's group relative to the South Asian British children's group. This is important as it relates to one of the key differences in responses between the two groups. That is, the higher likelihood of feeling embarrassed may be a reflection of the age difference between the groups (i.e. a larger proportion of the South Asian cohort being in the pre-adolescent age group), rather than the ethnic difference. This is an important confounder and should be addressed in the discussion. 3. At the editor's discretion, the COREQ 32-item checklist could be included as supplementary material.
--	---

VERSION 1 – AUTHOR RESPONSE

Reviewer #1

The redraft is much improved and its contribution to understanding the issues for health service design for children and young people from South Asian backgrounds is very and insights can be transferred to other service areas treating chronic conditions in childhood.

**** Many thanks ****

Reviewer #2

This manuscript is a significant improvement over the previous. All prior reviewer concerns have been well addressed. The inclusion of many examples of specific responses from the study participants enhances the interest and usefulness of this paper both for practicing clinicians and researchers.

I have only a few minor comments:

1. In the results section titled "Children's interactions with healthcare system" - line 239 states that the lack of recall of children regarding the time they were diagnosed with asthma "may reflect the quality of the interaction they had had with the HCP's." Is there any evidence to support this interpretation? I would suggest removing this statement unless there is further evidence within the study or published literature to support it.

Furthermore, this statement, as well as the statement in line 241, "suggesting a lack of recall may be age-and time-dependent" are interpretations and should be included in the discussion, rather than results section of the manuscript.

**** Many thanks for your suggestions. We have amended this in the following lines 250-251 and 386-387****

2. I note that there are significantly fewer 11 and 12 year olds (i.e. older children) among the White British children's group relative to the South Asian British children's group. This is important as it relates to one of the key differences in responses between the two groups. That is, the higher likelihood of feeling embarrassed may be a reflection of the age difference between the groups (i.e. a larger proportion of the South Asian cohort being in the pre-adolescent age group), rather than the ethnic difference. This is an important confounder and should be addressed in the discussion.

****Many thanks. We have now discussed this in Line 425-427****

3. At the editor's discretion, the COREQ 32-item checklist could be included as supplementary material.

**** This has been amended****

VERSION 2 – REVIEW

REVIEWER	Dhenuka Radhakrishnan Children's Hospital of Eastern Ontario, Ottawa, Ontario, Canada
REVIEW RETURNED	09-Nov-2018
GENERAL COMMENTS	All reviewer comments have been satisfactorily addressed.